# Fabrication of Hydrogel-Based Composite Fibers and Computer Simulation of the Filler Dynamics in the Composite Flow

**DOI:** 10.3390/bioengineering10040448

**Published:** 2023-04-06

**Authors:** Thomas Gruhn, Camilo Ortiz Monsalve, Claudia Müller, Susanne Heid, Aldo R. Boccaccini, Sahar Salehi

**Affiliations:** 1Department of Biomaterials, Faculty of Engineering Science, University of Bayreuth, Prof.-Rüdiger-Bormann Str. 1, 95447 Bayreuth, Germany; 2Invertec-eV, Gottlieb-Keim-Straße 60, 95448 Bayreuth, Germany; 3Institute of Biomaterials, Department of Materials Science and Engineering, University of Erlangen-Nuremberg, Cauerstraße 6, 91058 Erlangen, Germany

**Keywords:** composite filaments, anisotropy, microrods, wet spinning, computational simulation

## Abstract

Fibrous structures with anisotropic fillers as composites have found increasing interest in the field of biofabrication since they can mimic the extracellular matrix of anisotropic tissues such as skeletal muscle or nerve tissue. In the present work, the inclusion of anisotropic fillers in hydrogel-based filaments with an interpenetrating polymeric network (IPN) was evaluated and the dynamics of such fillers in the composite flow were analyzed using computational simulations. In the experimental part, microfabricated rods (200 and 400 μm length, 50 μm width) were used as anisotropic fillers in extrusion of composite filaments using two techniques of wet spinning and 3D printing. Hydrogels such as oxidized alginate (ADA) and methacrylated gelatin (GelMA) were used as matrices. In the computational simulation, a combination of computational fluid dynamics and coarse-grained molecular dynamics was used to study the dynamics of rod-like fillers in the flow field of a syringe. It showed that, during the extrusion process, microrods are far from being well aligned. Instead, many of them tumble on their way through the needle leading to a random orientation in the fiber which was confirmed experimentally.

## 1. Introduction

In the field of tissue engineering and biofabrication, hydrogels as polymeric matrices play an important role in mimicking the properties of the extracellular matrix (ECM) of soft tissues [1]. They contain more than 90% of water and form gel through various intermolecular attractions and covalent bonds and crosslinking of long polymer chains. Additionally, they can be loaded with cells for therapeutic applications [2,3,4,5,6,7]. Nevertheless, hydrogels have low mechanical properties and a fast degradation rate which can be adjusted by different crosslinking methods and the production of the blends and composites in combination with other materials [5,7,8]. For example, to improve the mechanical properties of hydrogels, recent studies have focused on developing an interpenetrating polymer network (IPN). They are formed by at least two incompatible polymers that after crosslinking result in a network of interlocking polymers and the structure will benefit from the properties of all components [3,6,9,10,11].

Alginate, a natural polysaccharide, can form hydrogels and be physically crosslinked. To improve its biodegradability, it could be oxidized, forming oxidized alginate also known as alginate dialdehyde (ADA) [12]. Moreover, to improve the biological interaction of living cells with alginate, it can be mixed with gelatin or its modified variants such as methacrylated gelatin (GelMA) [3,11]. Different groups have focused on the evaluation of IPNs formed by alginate (or ADA) and gelatin (or GelMA). Jeon et al. evaluated ADA–GelMA IPN’s mechanical and biological properties, demonstrating higher toughness, flexibility, and elasticity of IPN hydrogels with respect to alginate hydrogels [3]. In such hydrogels, two external crosslinking steps such as ionic crosslinking with CaCl_2_ solutions and photo crosslinking should be applied and there will be an internal crosslinking of macromolecules where aldehyde groups from ADA and amine groups from GelMA react via Schiff base reaction, forming an imine bond [3,13,14].

Depending on the type of target tissue to mimic and its isotropic or anisotropic properties, the structure and composition of the hydrogels can be defined [15]. In various tissues such as bone [16], or skeletal muscle [17], the anisotropic properties are given by the unique alignment and orientation of the collagen fibrils in the structure of the ECM which supports their particular functions, such as mechanical loading [18,19]. Furthermore, anisotropy is crucial for tissue functionality. For instance, in muscle sarcomere, actin filaments and myosin motor proteins rearrange anisotropically during the contraction process [20,21].

Fibrous scaffolds represent a strong candidate for mimicking the anisotropic tissue microenvironment by resembling morphologically the fibrous components of the ECM [22,23]. Furthermore, they have a high capacity for carrying substances such as drugs, biological factors, and cells [23,24,25]. Fibers can be produced through various fabrication techniques including polymer extrusion, fiber spinning, and 3D printing. In particular, wet spinning has been widely used for the fabrication of fibers derived from natural sources such as chitosan [26], collagen [22], and silk fibroin [27]. Briefly, wet spinning is a phase inversion technique in which a polymer solution is extruded in a coagulation bath, which does not dissolve the polymer but solidifies it [22]. The formed filament is taken from the bath and can be further processed by stretching or crosslinking. Multiple parameters in the wet spinning technique, including polymer/coagulation bath system, polymer concentration, nozzle diameter, feed rate, and winding velocity, affect the fiber’s properties and structure [28,29]. One of the main advantages of wet spinning is that filaments can be easily loaded with different sizes and types of fillers, as well as therapeutic substances [29,30]. One of the other techniques to process the hydrogels in the form of filaments is 3D printing which can also fabricate complex 3D structures. Three-dimensional printing of materials can be based on laser, injection, or extrusion. For example, in extrusion-based printers, through continuous extrusion, the filaments can be printed. In this technique, the temperature can be applied based on the type of polymer to melt and extrude the polymer, or hydrogels can be extruded based on the desired patterns [31,32,33].

To generate hydrogel-based composite fibers, fillers with various geometries and properties can be added to the matrix [34,35]. These fillers can have isotropic structures such as microspheres and microgels with an aspect ratio close to one, or anisotropic structures with rod-shaped morphology and high aspect ratios such as microcylinders, microribbons, and microrods generating heterogeneous constructs [36]. Furthermore, technologies such as fiber spinning [37] or 3D printing [38,39] can be used to produce filamentous materials with anisotropic properties. To address this, Prendergast et al. [39] have developed an approach to align hydrogel-based norbornene-modified hyaluronic acid short fibers (length ~20 µm) within 3D bioprinted filaments of GelMA, where applied shear stresses, during 3D printing, resulted in aligning those short fibers and photocrosslinking stabilized their orientation. Furthermore, Omidnia et al. [40] have reported the alignment of magneto-responsive poly (d,l-lactide-co-glycolide) (PLGA) short fragments (lengths ranging between 25 and 105 µm and diameters of 689.7 ± 88.5 nm) labeled by magnetic particles under the influence of the magnetic field within the hydrogel matrix. We also previously reported the fabrication of rod-shaped cell carriers composed of PLGA using a combination of micropatterning and spin coating, which were subsequently biofunctionalized and loaded with skeletal muscle cells. Such freestanding carriers could be fabricated in various lengths (100 to 500 µm) to be applied as injectable cell carriers to shield the adherent cells from mechanical shear forces applied by direct injection of the suspension [36].

In this study, we show the fabrication of composite filaments containing long microrods. To the best of our knowledge, fabrication of filaments with microrods fillers longer than 100 µm has not been reported before. The importance and potential application of such constructs are in situ generation of the anisotropic structures for biofabrication of tissues such as skeletal muscles which are made of long, multinucleated, and unidirectional myofibers (differentiated cells with length <500 µm). Therefore, as hydrogel matrices alone cannot provide adhering cells appropriate contact guidance, the insertion of microfillers with a higher aspect ratio can support the adhesion of myoblasts and their differentiation to the functional myofibers. Evaluation of optimal conditions regarding materials and processing for biofabrication approaches requires a significant experimental effort, which is typically an expensive and time-consuming process. As much as the inclusion of isotropic and anisotropic fillers in hydrogel matrices is becoming increasingly relevant in the field of tissue engineering, it is also feasible to use simulation tools to analyze the dynamics of these fillers within the flow [41]. Accurate control of orientations and alignments of the fillers would enable users to fabricate 3D composite fibrous structures with anisotropic material composition. In this study, we show how such microrods can be used in the formation of composite filaments and how applying a combination of computational fluid dynamics [42] (CFD) and coarse-grained molecular dynamics (MD) [43] can support the simulations and prediction of the flow dynamic and movement of the fillers during the extrusion. The use of numerical simulations represents a powerful tool for analyzing relevant parameters and predicting optimal conditions avoiding complex experimental processes. In this study, two sets of experimental designs will be reported that both aim at the production of composite filaments with rod-shaped fillers: (1) wet-spun composite filaments and (2) composite filaments extruded using a 3D printer. It should be noted that the simulations of the filler dynamics help to predict the orientational order of the fillers as they pass the syringe and the needle. The orientational distribution of fillers influences the anisotropy of the composite filaments, which modifies properties such as mechanical properties and thermal transport properties. We will show, with the help of the simulations, the orientation dynamics of fillers as they pass through the syringe and needle, depending distinctly on the filler length. Figure 1 and Appendix A show a schematic visualization of the microrods flowing through the syringe and the needle during the extrusion process which is the base of both fiber spinning techniques considered in this study. The rod dynamics and their alignment are controlled by the gel velocity field, indicated by the red lines, length, and density of the microrods (Figure 1).

## 2. Materials and Methods

### 2.1. Materials

List of the chemicals used in this study are as follows: Gelatin type A from porcine skin (G2500, Sigma Life Sciences, St. Louis, MO, USA), sodium alginate (VIVAPHARM^®^ alginate PH163 S2, from brown algae, with approval as a pharmaceutical excipient, JRS PHARMA GmbH & Co. KG, Rosenberg, Germany), methacrylic anhydride (MA, Sigma-Aldrich, St. Louis, MO, USA), Dulbecco’s Phosphate Buffered Saline (DPBS, Sigma-Life Science), poly (d, l-lactide-*co*-glycolide) (PLGA 75:25, 66,000–107,000, Sigma-Aldrich), polyvinyl alcohol, (PVA, 86–89% low molecular weight ThermoFisher, Kandel, Germany), calcium chloride dihydrate (CaCl_2_·2H_2_O, Carl Roth, Karlsruhe, Germany), dichloromethane (DCM, >99.8%, Sigma-Aldrich), sodium periodate (Carl Roth), ethanol absolute (VWR International), ethylene glycol (Carl Roth), and lithium phenyl-2,4,6-trimethyl-benzoyl phosphinate (LAP, Tokyo Chemical Industry (TCI), Tokyo, Japan).

### 2.2. Synthesis of GelMA and ADA

GelMA synthesis was performed following previous work published by Ebrahimi et al. [28]. Briefly, gelatin was dissolved in DPBS at 50 °C followed by the addition of methacrylic anhydride. After about 1 h, the methacrylation reaction was terminated by the addition of preheated DPBS. Next, the supernatant was dialyzed against warm milli-Q water for a week followed by lyophilization for a week. GelMA product was 80% methacrylated and stored at −20 °C for further use.

Alginate dialdehyde (ADA) was also synthesized following Heid et al. [44]. Alginate was dissolved in ethanol followed by the addition of dropwise sodium metaperiodate which was previously dissolved by adding 1.337 g in 50 mL MilliQ^®^ water. The reaction was left for 6 h in the dark and then quenched by adding ethylene-glycol. Next, the reaction was left to sediment for 5 min followed by ethanol subtraction. The resulting solution was dialyzed against MilliQ^®^ water for a week followed by lyophilization for another week. The final product had 11% oxidation degree and was stored at room temperature for further use.

### 2.3. Fabrication of Rod-Shaped Fillers

Microrods were prepared as described by Salehi et al. [36]. PLGA solution (12 mg/mL in DCM) was cast and spin-coated using a spin coater (KLM Spin-coater SCE-150, NOVOCONTROL Technologies GmbH & Co. KG, Montabaur, Germany) on the top of a polydimethylsiloxane (PDMS) mold with micropattern consisting of 50 μm width microgrooves and 200 and 400 μm length sizes. To remove the PLGA layer, 10 wt% PVA solution was cast as a sacrificial layer and left to dry overnight. Therefore, freestanding microrods were obtained by dissolving the PVA layer in water. The estimated rod’s concentration was obtained by counting the number of them as a suspension in water using optical microscopy (DMI3000B, Leica Microsystems Ltd., Wetzlar, Germany).

### 2.4. Wet Spinning of ADA-GelMA Filaments with Rod-Shaped Fillers

Filaments were produced by the wet spinning of a blend solution of ADA and GelMA (ADA–GelMA) with a ratio of 2:1 and a final concentration of 10 wt%. The blend solution was prepared after dissolving the lyophilized ADA and GelMA in DPBS at 50 °C for 3 h. An amount of 1 wt% LAP was added to the blend solution as a photoinitiator and 14,000 rods/mL with lengths of 200 and 400 μm were mixed with the hydrogel-based spinning solution. The ADA–GelMA blend hydrogel was further launched in a syringe pump (Harvard Apparatus, South Natick, MA, USA) using a 10 mL syringe and extruded with different feed/flow rates of 50, 100, 200, and 400 (μL/min) through metallic needles with inner diameters of 0.4 mm (27 G) and 0.8 mm (21 G) (Blunt, B. Braun, Melsungen, Malaysia) into a CaCl_2_.2H_2_O solution as a coagulation bath with 2 and 4.5 wt%. Filaments were submerged into the bath for 5 min for ionic crosslinking of ADA and after being taken out from the bath using forceps were exposed to visible light (400–500 nm wavelength) for 1 min for photocrosslinking. The inclusion of rods was evaluated by optical microscopy. The alignment of rods within the filaments was calculated with respect to the fiber axis using ImageJ software. (A total of 70 rods were analyzed for each size.)

### 2.5. Viscosity Measurement of the Spinning Solution

We further measured the viscosity of the ADA-GelMA hydrogel solutions using a rheometer (Discovery HR-2, TA Instruments, New Castle, DE, USA) with cone–plate geometry and a flow sweep from 0.01 to 1000 1/s at 21 °C. For hydrogel solutions with fillers, plate–plate geometry was used, and a flow sweep was measured from 0.1 to 100 1/s at 21 °C.

### 2.6. Extrusion of GelMA Hydrogel with Rod-Shaped Fillers Using a 3D Printer

To prepare the composite ink for a 3D printer, the 10 wt% lyophilized GelMA hydrogel was dissolved in DPBS at 37 °C, and 14,000 rods/mL were added to the hydrogel followed by filling the cartridge with it. Similarly, rods with a length of 200 and 400 μm and a width of 50 μm were used. The composite ink was mixed and cooled down for 20 min to room temperature (20.5 °C) under a dynamic shaker (rotating 180° at a rate of 13 rpm) to avoid the sedimentation of the rods before printing. Continuous parallel filaments were extruded using a 3D printer 3D Discovery extrusion-based bioprinter (RegenHU, Switzerland), and G code was created using the software BioCAD (RegenHU, Switzerland). The size of the design was 30 × 6 mm. The air pressure was varied between 0.1 and 0.3 bar and the nozzle moving speed was between 5 mm/s in order to obtain the best printing conditions. The nozzle diameter was 0.41 mm (22G, Drifton, Hvidovre Dinamarca). The distance between the nozzle tip and the substrate was kept at 0.43 mm. To evaluate the alignment, images were acquired using optical microscopy (Leica DMi8, Germany). The angle between the horizontal axis and each rod was measured using FIJI software (version 1.52p, NIH, Bethesda, MA, USA). If the angle was between 0° and 20°, the rod was considered aligned. A total of 70 rods were analyzed for each size.

### 2.7. Statistical Analysis

All results are presented as mean ± standard deviation (SD). All the values were averaged at least in triplicate and statistical analyses were performed using Origin 2019b software (OriginLab, Northampton, MA, USA). A value of *p* < 0.05 was considered statistically significant.

## 3. Computational Simulation

### 3.1. Simulation of the Composite Flow

The dynamics of fillers as they flow through the syringe and needle were studied numerically. In the past, various methods have been used to investigate the flow of a rod-like particle in a viscous fluid. Examples are methods based on Brownian dynamics, the reciprocal theorem, the grand mobility matrix, or smoothed particle hydrodynamics [45,46,47,48]. Here, we used a combination of computational fluid dynamics [49] (CFD) and coarse-grained molecular dynamics (MD) simulations [43]. Fluid mechanics were determined by solving the Navier–Stokes equations for an incompressible, non-Newtonian fluid, namely Equations (1) and (2).
(1)$$\rho \left(\frac{\partial u}{\partial t}+\left(u\cdot \nabla \right)u\right)=-\nabla p+\nabla \cdot \left(\mu \nabla u\right)$$
∇ · u = 0(2)
with a flow velocity u(r,t) of the hydrogel at position r and time t. The mass density of the hydrogel is ρ, μ is the shear-dependent viscosity, and p(r,t) is the pressure field. Equation (2) ensures incompressibility.

The dependence of the viscosity μ on the shear rate $\overset{\cdot }{\gamma }$ was modeled with the power law equation:(3)$$\mu =m\cdot {\overset{\cdot }{\gamma }}^{n-1},$$
where the consistency index m and the flow behavior index n were fitted to the experimentally measured curves of viscosity as a function of shear rate. We found the values m = 5.84 ± 0.77 Pa·s and n = 0.484 ± 0.04.

With the help of the software OpenFoam [42], the equations were solved numerically for the geometry of the syringe and needle used in the wet spinning system (Figure 2A). In a similar way, Emmermacher and coworkers have studied analytically and with CFD the flow behavior of a bioink through an extruder [50]. Due to the radial symmetry of the flow field, the fluid dynamics calculations could be carried out in two dimensions, resembling a thin wedge with the symmetry axis of the system at the bottom and the upper boundary following the size of the radius in the syringe and the needle, respectively (Figure 2B). Details are given in Table 1. For the diameter of the needle, we considered two cases, d_n_ = 0.4 mm and d_n_ = 0.8 mm.

The flow field was calculated numerically with a mesh that was refined in regions of stronger velocity changes (Figure 2C). The accuracy of the calculation was checked by comparison with higher mesh resolutions.

The system was studied for flow rates of 50, 100, and 500 μL/min. In the calculations, fixed pressure values were chosen at the inlet and the outlet. These conditions led to much higher numerical stability in comparison to calculations with a fixed flow rate at the boundaries. Atmospheric pressure was chosen at the outlet while individual runs were performed with different inlet pressure. Iteratively, the inlet pressure was varied until the pressure difference led to one of the desired flow rates. The flow rate was determined by integrating the velocity over the outlet cross-sectional area. We used no-slip boundary conditions for the velocity field (∇v= 0) and zero von Neumann boundary conditions for the pressure field ($e_{⊥}\cdot \nabla P$ = 0 with a unit vector $e_{⊥}$ normal to the surface).

After obtaining the steady state velocity profile, the axial and radial components of the velocity field were fitted by analytic functions. Syringe and needle geometry were split into smaller regions, in which the velocity could be fitted by quadratic functions of the axial and radial coordinates. This way, the deviation between the numerical values and the fit functions was less than 5% in all regions.

### 3.2. Simulation of the Filler Dynamics in the Composite Flow

With the obtained flow field, the flow behavior of microrods was simulated using coarse-grained molecular dynamics. Microrods were modeled as bonded spherical particles linked through elastic bonds. The number of particles per microrod depended on the real rod length (Figure 3).

The dynamics of the rods were studied in a 3D system, using the calculated flow field with cylindrical symmetry. The motion of each particle is calculated by numerically solving the dynamic equations. For particle i, one has:(4)$$m_{i}{\ddot{r}}_{i}=\sum_{j\neq i}f_{ij}^{p}+\sum_{j\;ngb}f_{ij}^{b}+\sum_{j,k\;ngb}f_{ijk}^{\vartheta }+\sum_{m=1}^{N_{w}}f_{im}^{w}+f_{i}^{h}$$
with the interaction force $f_{ij}^{p}$ caused by particle j, the bond force $f_{ij}^{b}$ caused by a bond with neighbor particle k inside the microrod, and a force $f_{ijk}^{\vartheta }$ that forces the angle $\vartheta \left(r_{ij},r_{ik}\right)$ between bond vectors $r_{ij}$ and $r_{ik}$ to be close to $\vartheta_{0}$. Furthermore, a force $f_{im}^{w}$ represented the interaction with wall m and a force $f_{i}^{h}$ inferred by the local flow field of the hydrogel. The first four types of forces are related to potentials,
(5)$$\begin{array}{ll} f_{ij}^{p}=-\nabla \Phi^{p}\left(r_{ij}\right), & f_{ij}^{b}=-\nabla \Phi^{b}\left(r_{ij}\right),\\ f_{ijk}^{a}=-\nabla \Phi^{a}\left(\vartheta \left(r_{ij},r_{ik}\right)\right), & f_{im}^{w}=-\nabla \Phi^{w}\left(r_{im}\right) \end{array}$$
which depends on the particle–particle distance $r_{ij}$, the particle wall distance $r_{im}$, or the angle between two bonds.

For the particle–particle interaction potential and the wall potential, we used the Week–Chandler–Anderson (WCA) potential and the corresponding wall potential, respectively [51].
(6a)$$\phi^{P}\left(r\right)=\left\{\begin{array}{cc} 4\varepsilon \left[{\left(\frac{\sigma }{r}\right)}^{12}-{\left(\frac{\sigma }{r}\right)}^{6}\right], & r\leq 2^{1/6}\sigma \\ 0, & r>2^{1/6}\sigma \end{array}\right.$$
(6b)$$\phi^{w}\left(r\right)=\left\{\begin{array}{cc} \frac{3\sqrt{3}}{2}\varepsilon \left[{\left(\frac{\sigma }{r}\right)}^{9}-{\left(\frac{\sigma }{r}\right)}^{3}\right], & r\leq 3^{1/6}\sigma \\ 0, & r>3^{1/6}\sigma \end{array}\right.$$

We used an interaction parameter $\varepsilon =6.8\cdot 10^{-7}J$ and a particle diameter of σ = 50 µm.

Harmonic interaction potentials were used for the bond and the angular potentials:(7a)$$\Phi^{b}\left(r\right)=K_{b}{\left(r-r_{0}\right)}^{2}$$
(7b)$$\Phi^{a}\left(r\right)=K_{a}{\left(\vartheta -\vartheta_{0}\right)}^{2}$$
where $r_{0}$ is the equilibrium distance. We used short and longer bonds between the nearest and next nearest particles in the microrod. They had an equilibrium length of r_0_ = 50 µm and r_0_ = 70.71 µm, respectively. In both bond potentials, the bond stiffness was $K_{b}=0.68\frac{J}{{\mu m}^{2}}$. The equilibrium angle between short bonds was $\vartheta_{0}=90{}^{\circ}$, while the equilibrium angle between short and longer bonds was $\vartheta_{0}=45{}^{\circ}$. A constant of K_a_ = 0.068 J was used in all cases.

The rod dynamics were studied in an NVT ensemble with a constant number of particles, temperature, and volume. For this purpose, the equations of motion of the particles are enhanced, according to the Nose–Hoover algorithm [43].

The drag force $f_{i}^{h}$ on the particle i is proportional to the difference between the local flow field $u(r_{i})$ and the velocity $v_{i}$ of the particle:(8)$$f_{i}^{h}=K(u(r_{i})-v_{i})$$
where K is the friction coefficient. In our model, we assume that the flow field changes only at a short distance to spheres, such that the drag force can be applied separately to all particles. This is a reasonable approximation for the given solvent. For these simulations, microrods with two different lengths of 200 µm and 400 µm and rod density of 0.1 mm^−3^ were used. In the beginning, rods of random orientational and spatial distribution were set into the broad, initial region of the syringe. Each setup was studied in 10 independent runs with different random initial configurations. The local orientational order of rods was determined by calculating the orientation tensor A with components:(9)$$A_{µ\nu }=\langle n_{µ}n_{\nu }\rangle =\frac{1}{N}\sum_{k=1}^{N}n_{µ}\left[k\right]n_{\nu }\left[k\right].$$
with n[k] being the unit vector parallel to the main axis of rod k. The scalar order parameter:(10)$$S=\frac{3}{2}A_{11}-\frac{1}{2}$$
determines the alignment of the rods in the direction of the symmetry axis of the needle. For a perfectly aligned set of rods, one has S = 1, while for completely random orientational distributions, the order parameter is S = 0 [52].

## 4. Results and Discussion

### 4.1. Microrod Productions

Microrods were fabricated using the microfabrication technique, and after removing the PVA layer, the freestanding rods were imaged using optical microscopy. Figure 4A,B show the morphology of microrods with 200 μm and 400 μm lengths dispersed in water, respectively.

### 4.2. Wet Spinning of the Filaments

The wet spinning of pure filament was first established and spinning parameters were set. The alginate part of the filaments was first ionically crosslinked in the CaCl_2_·2H_2_O bath, and further, the filaments were photocrosslinked using the visible light. It has been already shown in the literature that the presence of a higher amount of alginate in comparison to GelMA [10] gives more stability to the fiber formation; therefore, we used a 2:1 ratio blend of ADA–GelMA. For the ADA–GelMA 2:1 ratio, stable and uniform filaments with an average diameter of ~380 μm were formed. Figure 4C shows the influence of the feed rate and needle diameter on the filament diameter when keeping the ADA–GelMA ratio (2:1), the hydrogel concentration (10 wt%), and the CaCl_2_·2H_2_O bath concentration (4.5 wt%) constant. After measuring the fiber diameters using optical microscopy, it is shown that with increasing the feed rate, the diameter of filaments has increased, and this is the case for both sizes of the needle. This result is in agreement with the literature, as Lee et al. after spinning the alginate filaments reported a similar trend [53].

Furthermore, composite filaments of ADA–GelMA containing rod-shaped fillers were fabricated using the same parameters as pure filaments. As shown in Figure 5, filaments containing microrods with varied lengths of 200 and 400 µm show uniform morphology and homogenous distribution. Microrods with a length of 200 µm preserved their shape; however, microrods with a length of 400 µm mostly bent and deformed in the filaments. This can be due to the tumbling behavior of microrods, which will be discussed further in the computational part, and the flexibility of the microrods after fabrication using a spin coater. It should be noted that rod-shaped fillers fabricated by the spin coating technique will have a thickness below 500 nm which is important for their application as a cell carrier [36]. In our previous study, we showed that the flexibility of the microrods (length < 500 µm) can shield the adherent muscle cells on the carriers from the mechanical stresses applied during the extrusion [36]. We showed that the viability of the cells was significantly higher than those cells which were directly injected as a cell suspension and the functionality of the adherent cells on the carriers was preserved after injection, where they were still able to differentiate and form long myotubes. Such tumbling of rod-shaped fillers has been also previously reported and simulated by Rubio et al. [54] where they reported nanorods (length and diameter of approximately 2 μm and 400–500 nm, respectively) tumble within the channel and in the flow of the liquid.

The angular distribution of microrods with 200 and 400 µm length with respect to the fiber axis was also measured and shown in Figure 5E,F, in which no specific alignment is observed and indicates that microrods are actually randomly distributed inside the filaments.

Fabrication of composite filaments containing anisotropic fillers such as fiber fragments has been studied before, too. Previous studies mostly evaluated the presence of fillers shorter than 100 µm within the extruded filaments and, for example, with the help of external forces tried to increase the alignment. Rose et al. reported the enhancement of the alignment of the microrods after applying external magnetic fields in the milliTesla order [41,55]. Furthermore, the previous studies showed the shorter the length of the rods, the higher the alignment during the extrusion will be [39,56,57,58]. However, such short fragments can only support the adherence of single cells.

To evaluate the effect of the microrods on the viscosity of the blend solution, the viscosity of the spinning solution was measured with and without the microrods at room temperature. Figure 6 shows the variation of shear rate (1/s) vs. viscosity (Pa∙s) for ADA–GelMA (10 wt%, 2:1) with and without microrods. As expected, the hydrogel composite exhibited a shear-thinning behavior since the viscosity decreases as the shear rate increases [14,54,55]. Furthermore, power law parameters were calculated as mentioned in Equation (3), being 5.84 (Pa∙s) ± 0.77 and 0.484 ± 0.04 for m and n, respectively. These values have been already reported for 1.5 wt% alginate (m = 0.346 Pa∙s and n = 0.840, respectively) [59]. In comparison, Paxton et al. calculated power law parameters for m and n equal to 13.3 (Pa∙s) and 0.608 for alginate–gelatin (4–20 wt%) [60]. Additionally, Distler et al. calculated an m value of 19.878 (Pa∙s) and an n equal to 0.225 for ADA–gelatin (3.75–7.5 wt%) [14]. Comparing these values shows that for alginate with a concentration between 1.5 and 8 wt%, an increase in concentration implies an increase in viscosity, as well as a higher deviation from the Newtonian behavior (values of n close to 1 reflect Newtonian flow) [58]. Moreover, results found by Paxton et al. and Distler et al. reflect a decrease in viscosity by increasing the shear rate and 0 < n < 1, an apparent non-Newtonian behavior [14,58]. This result is consistent with that found in the present study, in which the coefficient of 0.484 reflects non-Newtonian behavior.

Similarly, the effect of microrods on the viscosity of the spinning solution was evaluated at room temperature which was the spinning temperature. Figure 6 presents the effect of the addition of microrods (200 μm length) on the viscosity of the composite hydrogel. The results showed a slightly higher viscosity of the blend hydrogel in the presence of the microrods compared to the control (ADA–GelMA, 10 wt%, 2:1). It is known that the viscosity of a polymer varies with respect to the aspect ratio of the filler. Thus, increasing the aspect ratio increases the surfaces which are interacting with the fluid and therefore increases the viscosity of the composite [37,39]. For rod-shaped fillers, the aspect ratio is much greater than one and increases as the length-to-diameter ratio increases [59]. Therefore, as expected, we observed a minimal increase in the viscosity of the hydrogel composite upon the addition of the microrods to the spinning solution.

Such behavior has been reported previously by Prendergast et al. [39], where they also observed a slight increase in the viscosity after the addition of about 3% of fiber fragments to the GelMA hydrogel. Sonnleitner et al. [37] also extensively studied the effect of various volume ratios of short fragments on hydrogel matrices and from the shear rheological analysis have concluded that short fibers such as polycaprolactone (PCL) and cellulose nanofibers within alginate and pluronic matrices can vary the viscosity of the hydrogel. Increasing the concentrations of both fillers resulted in increasing the viscosity. This observation was interpreted with the surface area of the fillers. Kosik-Kozioł et al. [57] have also demonstrated that the alginate ink with and without the poly lactic acid (PLA) short fibers showed shear-thinning behavior. They have also reported the higher zero-shear rate viscosity of the alginate in the presence of 1% PLA sub-micron fiber (from 26 to 35 Pa s). They discussed that the viscosity of the suspension was affected by the Brownian motion of sub-micron fibers and the movements of the macromolecular chains that move around each other for overall flow to happen. These interactions result in high resistance and therefore in a higher measured viscosity.

### 4.3. Extruded Filaments Using a 3D Printer

The composite filaments containing the microrods were further extruded using a 3D printer. They were composed of GelMA hydrogel (10 wt%) including rod-shaped fillers. The composite hydrogel after adjusting the temperature to 20.5 °C was extruded in the form of parallel filaments using a 3D printer (Figure 7A). Figure 7B,C show the microscopical images of the filaments containing microrods with a length of 200 and 400 μm. The morphology of the printed filaments due to the temperature sensitivity of the GelMA is inhomogeneous but the rods are visible within the strands. After quantification of alignment of the microrods with respect to the axis of the extruded filaments, it can be seen that the microrods with lengths of 400 µm were more aligned in comparison to the 200 µm length rod-shaped fillers. Printing of composite inks containing fillers with high aspect ratio has been recently more in focus as it concerns the direct fabrication of the anisotropic structures with oriented fillers to enhance the mechanical properties or alignment of the encapsulated cells within the ink for the field of biofabrication [39,40,56,61]. For example, Prendergast et al. [39] reported 3D printing of short filaments made by electrospinning technique with an average length of 20 µm and an average diameter of 5 µm (aspect ratio of 4) within the GelMA matrix. They investigated the effect of various printing parameters on the fabrication of hydrogel filaments containing aligned short filaments via extrusion printing. They also showed that shear stresses during printing enhance the alignment of the short filaments within the ink [39]. Fibers used in this study from the aspect ratio were similar to the system used in our current study.

### 4.4. Computational Simulation

We studied the polymer–filler system with numerical methods described in Section 3.1. The approach can be assigned to the class of unresolved CFD–DEM simulations that combine computational fluid dynamics (CFD) with discrete element methods (DEM) [62,63,64]. Different expressions based on hydrodynamic forces are used for coupling the solid fillers with the fluid matrix [64].

### 4.5. Flow Simulation

As mentioned above, the inlet pressure was iterated until the desired feed/flow rate was found. The final pressures for each configuration are reported in Table 2. Figure 8A shows the flow rate at the outlet over time, for the selected pressures (hydrogel density 1030.68 kg/m^3^, as in the experiment). The plot shows the increase in flow rate until the steady state with the desired flow/feed rate is reached (50, 100, and 500 μL/min). The results are shown for a needle size of 0.8 mm.

The amplitude of the flow rate along the geometry and streamlines of the velocity field is shown in Figure 8B,C. As expected, the flow velocity is lowest in the syringe region with the largest diameter and is maximum at the symmetry axis of the needle.

The flow field over radial distance R from the symmetry axis is shown in Figure 9A for different flow rates and a constant flow rate of 100 μL/min and different needle diameters (Figure 9B). In all cases, the velocity profiles are in excellent agreement with the analytic solution for the steady state velocity distribution of a non-Newtonian power law fluid in an infinite cylinder. The ratio between the diameter of the syringe and that of the needle is approximately 17.7. The linear flow velocity in the center of the needle is proportional to the inverse of the cross-sectional area. Therefore, it is about 300 times higher in the needle than in the syringe. A similar geometry was analyzed by Aguado et al. [65], who simulated the shear rate of alginate flow through a syringe–needle system with a diameter ratio of approximately 17. In their study, they found that the velocity in the needle is approximately 294 times higher than in the syringe. In addition, their findings showed that the shear stress increases considerably in the constriction region and, especially in the needle range, it is much higher close to the wall; the shear stress close to the wall is much more pronounced for the non-Newtonian fluid (alginate) than the shear stress at the wall of a cylindrical pipe in a Newtonian fluid (PBS) [65]. The difference in shear stress is due to the shear-thinning that is characteristic of the polymers [66]; the velocity profile is consistent with the findings of the present study (Figure 9).

### 4.6. Filler Dynamics

As mentioned above, the coupling of the steady-state flow and the molecular dynamics simulation was conducted using the drag force defined in (Equation (10)). The inertia part of the equations of motion is small so that the centers of mass of the microrods follow the streamlines. Figure 10A shows a superposition of microrods (200 μm) as they flow along the streamlines.

Figure 10B shows the order parameter S along the syringe and needle geometry for 200 μm microrods. The results are shown for different flow rates (50, 100, and 500 μL/min), using a needle diameter of 0.8 mm. The rods are positioned with an isotropic orientation distribution in the syringe in a range of 0 to 6 mm on a scale along the symmetry axis. Thus, the order parameter S is close to zero in this region. In the conical geometry of the syringe, the order parameter increases above 0.8 and decreases as the rods enter the cylindrical part with a smaller diameter. Figure 10B shows that the flow rates have no strong influence on the spatial distribution of the order parameter. This is related to the fact that the inertial part of the equations of motion plays a minor role. After the second conical region, at the beginning of the needle, the order parameter shows another minimum until it returns to a value of S ≈ 0.6 where it remains all along the path through the needle. In Figure 10C, the orientational order is compared for microrods with 200 µm and 400 μm lengths at a flow rate of 100 μL/min. For the longer microrods, the order parameter shows a distinct oscillation with a fixed wavelength of about 7.1 mm, while smaller peaks are observed for 200 μm length microrods. The oscillations are related to a tumbling of the microrods, comparable to the Jeffery orbits of rods in a constant shear flow [67]. It is remarkable that the tumbling is spatially synchronized such that the orientational order parameter, sampled over all rods, has minima at regular distances along the needle direction, even though the shear rate varies with the radial distance from the symmetry axis. Note that in the center of the rod, the shear rate is zero and the rods do not tumble at all (Figure 11 and Appendix A).

For both studied rod lengths, the order parameter indicates an orientational order that is far below perfect alignment (S = 1). This phenomenon is reflected in the microscopical images (Figure 5 and Figure 7) that show rods with low orientational order in the extruded filaments.

Multiple authors have simulated rod-shaped fillers through different flows. Kim et al. [68] simulated the flow of nanowires (100 nm diameter, 20 μm length) embedded in a polymer matrix through a nozzle with a circular cross-sectional area (840 μm diameter, 17 mm length), and one with an elliptical cross-sectional area (tapered end part of 3 mm, and diameters of 1.2 mm and 360 μm along the ellipsoid main axes). In their findings, they mentioned a good alignment for both geometries [68]. These results differ from those found in the present work. One important difference between both models is the aspect ratio of the rods, which is close to 200 for Kim et al. [68] and about 4 for the present work. This large difference in aspect ratio could explain the differences in alignment. Similarly, Lewicki et al. [69] found good alignment for carbon filaments with an aspect ratio higher than 50. On the other hand, Rubio et al. [54] evaluated the flow of cylindrically shaped rods with an aspect ratio of around 4 through a microchannel with a rectangular cross-sectional area and cone-like nozzle. They found that the nanorods tumble along the microchannel while for the nozzle outlet they are quite aligned [54]. This result is consistent with what was found in our present study, as the microrods rotate along the nozzle, but are highly aligned in the conical regions of the syringe (Figure 10B,C).

The mentioned aspects lead to the conclusion that a good strategy to increase the alignment of the microrods would be either to use a conical needle geometry or to increase the aspect ratio either by increasing the length or decreasing the diameter of the microrods.

## 5. Conclusions

In the present work, the inclusion of anisotropic (microrods) fillers in hydrogel-based wet-spun and 3D-printed filaments was evaluated. Microrods with 200 and 400 μm length and 50 μm width were distributed homogenously within the extruded filaments. Likewise, the dynamics of these fillers in the composite flow were analyzed by means of computational simulation. We could successfully develop a simulation system that represented the experimental part and the filler dynamics. Thus, by using real parameters such as the geometry of the system and the power law viscosity constants, it was found that the orientation of the microrods varied significantly along the geometry. Moreover, the microrods tumble during flow through the needle, and therefore, after the extrusion, their orientation in the fiber can vary significantly. This result is consistent with what was found experimentally, in which no preferential orientation of the microrods with respect to the axial length of the filament was found. The model used in the present work can be slightly varied to evaluate multiple parameters such as different geometries, filler length, flow velocities, etc. In future work, we will focus further on various concentrations of hydrogels from less to high viscous flow to investigate their effect on the rotational dynamics of the fillers. Furthermore, we will vary the concentrations of the fillers to higher numbers to evaluate their interaction in flow and their effect on the viscosity and mechanical properties of the filaments. For the simulation part, we will evaluate the flow of the fillers under the effect of the other types of geometries such as conical needles to investigate parameters that could efficiently increase the alignment of the microrods.

## Figures and Tables

**Figure 1 bioengineering-10-00448-f001:**
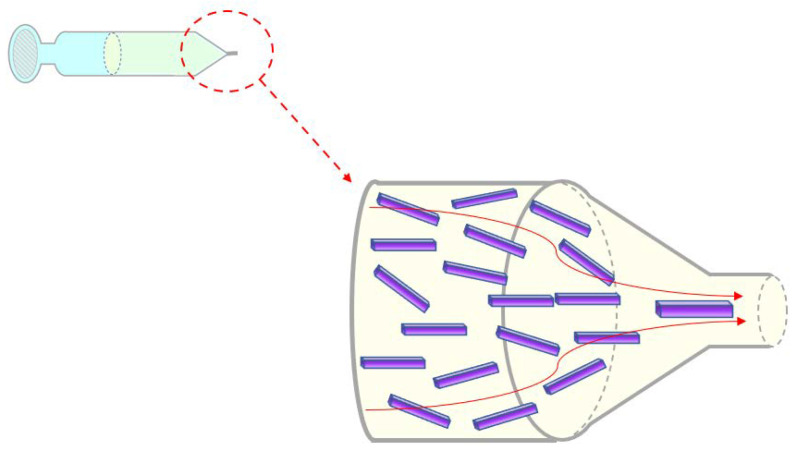
Microrods flow and alignment through the extruder-like geometry are affected by hydrogel matrix flow, length, and density of microrods.

**Figure 2 bioengineering-10-00448-f002:**
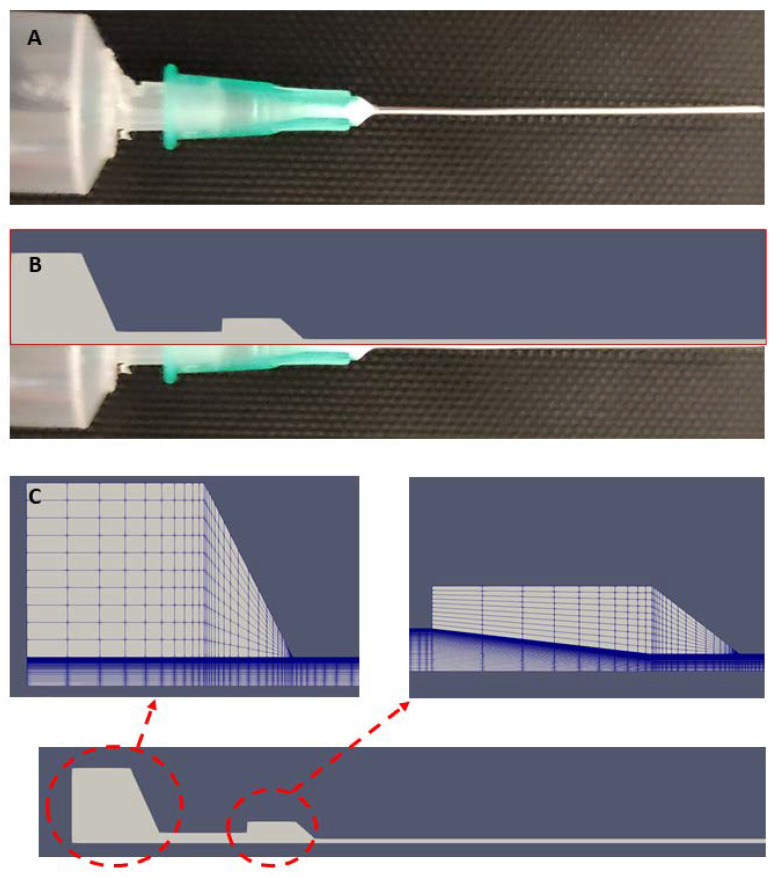
(**A**) Wet spinning syringe and needle geometry in the experiment. (**B**) The wedge geometry is designed using the internal dimensions of the experimental system. (**C**) Schematic of the wedge geometry and the mesh used in the CFD calculations.

**Figure 3 bioengineering-10-00448-f003:**
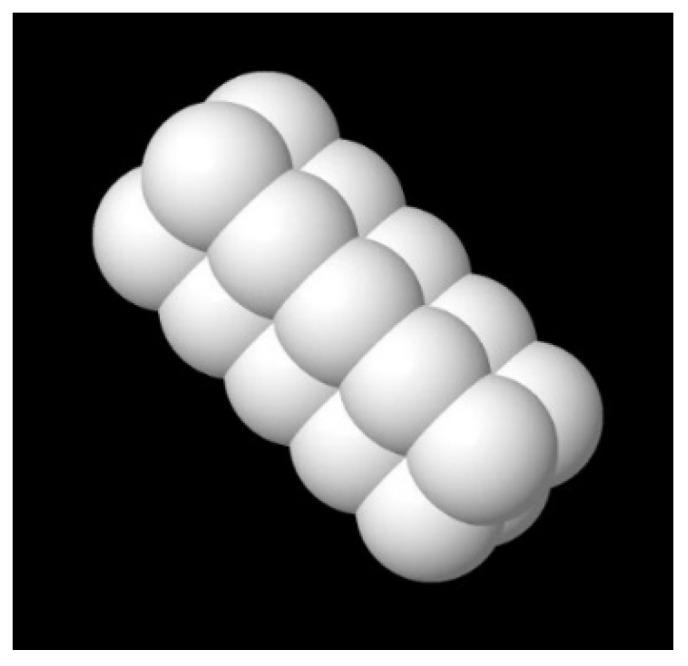
Model of the microrod as a group of bonded particles linked through elastic bonds.

**Figure 4 bioengineering-10-00448-f004:**
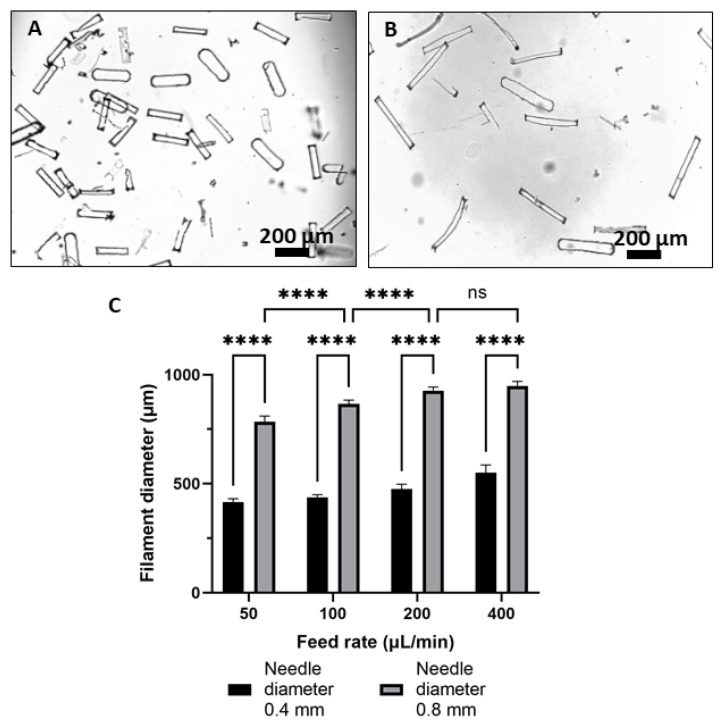
Optical micrographs presenting the morphology of freestanding microrods suspended in water, (**A**) 200 µm length and (**B**) 400 µm length, and (**C**) variation of the diameter of wet spun ADA–GelMA filaments based on using the different needle diameter and feed rate (****, *p* < 0.0001).

**Figure 5 bioengineering-10-00448-f005:**
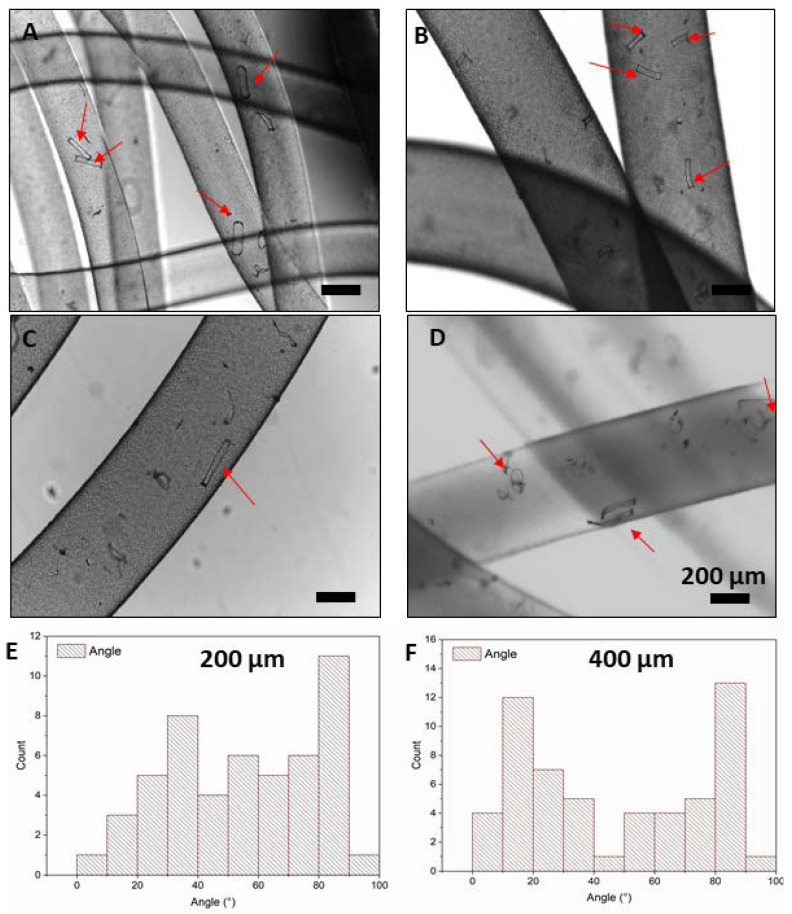
Images taken by optical microscopy show the morphology of the hydrogel-based filaments (ADA-GelMA, 2:1, 10 wt%) with rod-shaped fillers of (**A**,**B**) 200 and (**C**,**D**) 400 μm length. These pictures are taken from different spots of the spun filaments. The red arrows spot the rods within the wet spun filaments. Angular distribution for the (**E**) 200 μm and (**F**) 400 µm length microrods (14,000 rods/mL) with respect to the fiber axis.

**Figure 6 bioengineering-10-00448-f006:**
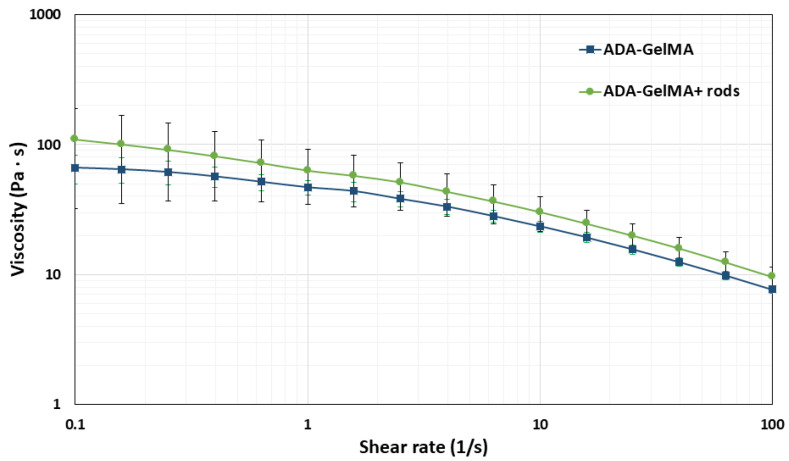
Shear rate (1/s) vs. viscosity for ADA–GelMA blend solution (10 wt%, 2:1), with microrods 200 µm length and without.

**Figure 7 bioengineering-10-00448-f007:**
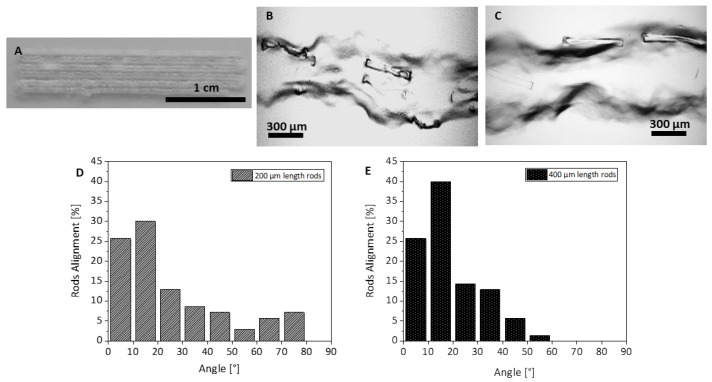
Morphology of the GelMA ink containing rod-shaped fillers extruded using a 3D printer: (**A**) macroscopic view of the parallel filaments, extruded strand containing microrods with (**B**) 200 µm length, and (**C**) 400 µm length. (**D**,**E**) Quantitative analysis of the aligned microrods in the hydrogel-based ink.

**Figure 8 bioengineering-10-00448-f008:**
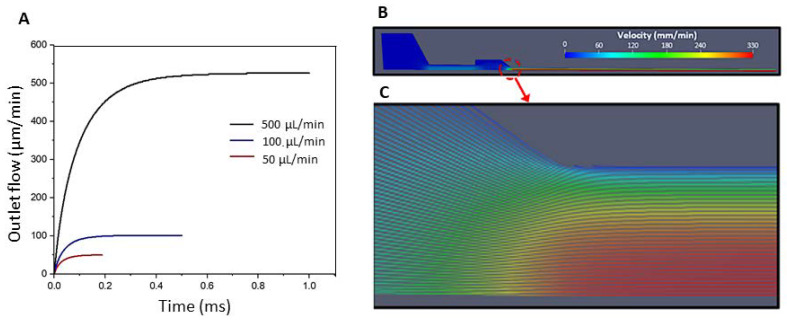
(**A**) Outflow rates for kinematic inlet pressures 6.4, 9, and 20 m^2^/s^2^ led to steady state flow rates of 50, 100, and 500 μL/min, respectively. (**B**,**C**) Variation of the velocity magnitude along the geometry. (**B**) Velocity profile for a feed rate of 100 μL/min and needle size 0.8 mm. (**C**) Streamlines.

**Figure 9 bioengineering-10-00448-f009:**
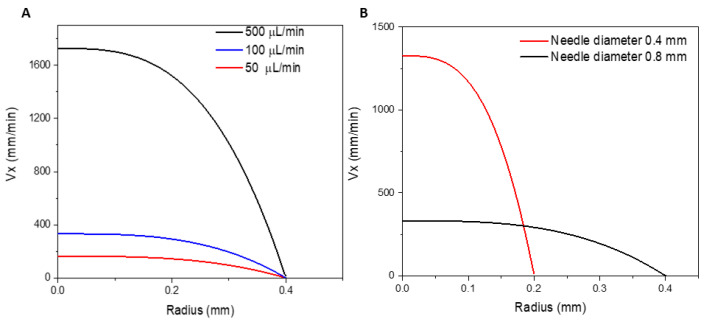
Variation of Vx profile at the outlet for different velocities with respect to radius evaluated at (**A**) different flow rates and at (**B**) different needle diameters and fixed flow rate of 100 µL/min. Vx is the horizontal x component of the flow velocity. The *x*-axis is parallel to the symmetry axis of the needle.

**Figure 10 bioengineering-10-00448-f010:**
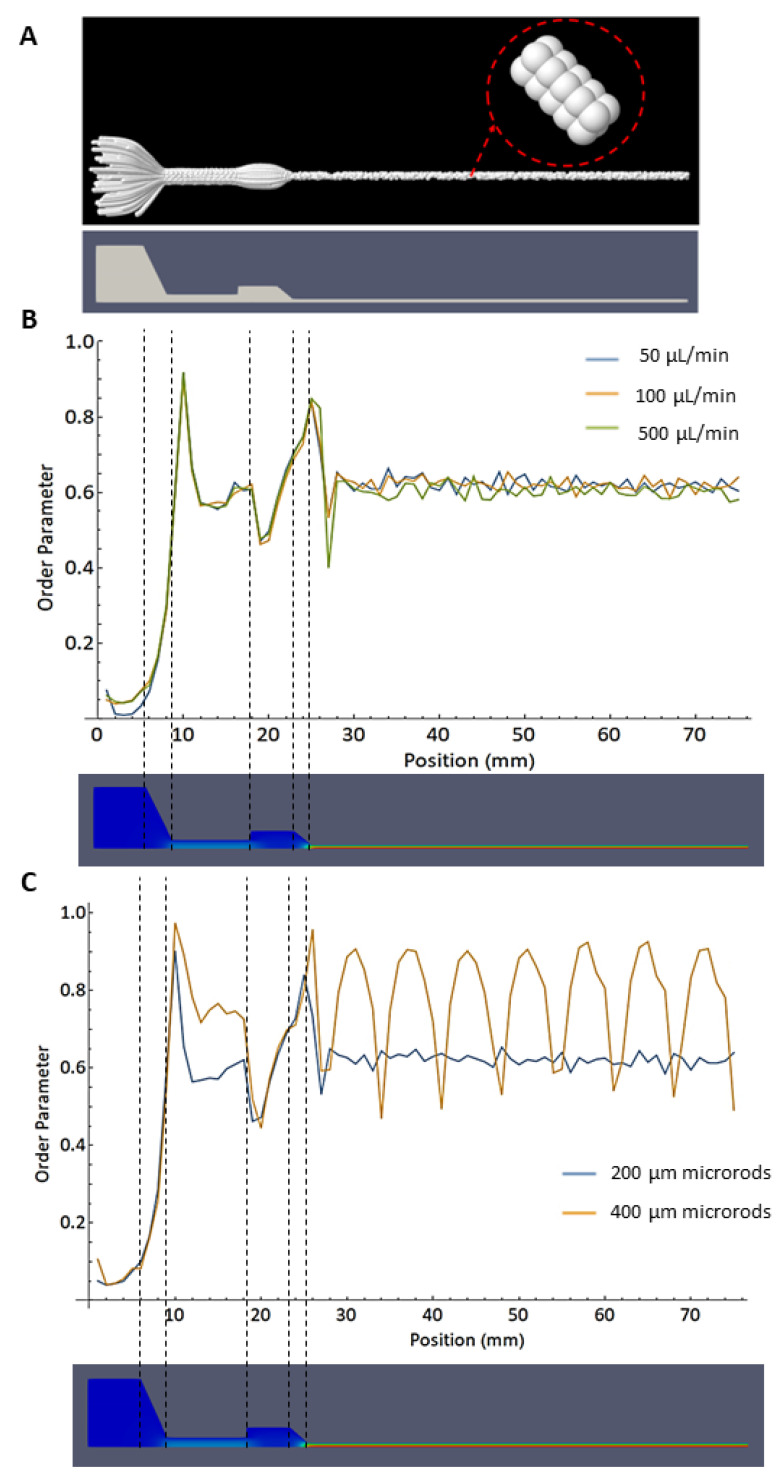
(**A**) Simulation with 100 microrods of 200 μm length. Multiple time steps are overlapped to observe the trajectory they follow. Variations of the orientational order parameter S (see Equation (10)) along the geometry, calculated for (**B**) 50, 100, and 500 μL/min flow rate (constant parameters: 200 μm microrods, needle diameter 0.8 mm) and (**C**) microrod’s lengths of 200 and 400 μm (constant parameter: 100 μL/min flow rate). For comparison, the contour of the syringe and needle is shown below.

**Figure 11 bioengineering-10-00448-f011:**
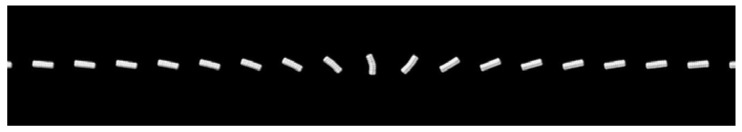
Microrods of 400 μm length rotating along the needle (needle size 0.8 mm, flow rate: 100 μL/min).

**Table 1 bioengineering-10-00448-t001:** Wedge geometry specifications.

Geometry Section	Distance, x (mm)	Diameter (mm)
Syringe (container)	0–6	14.14
Syringe cone	6–9	from 14.14 to 2
Syringe (junction)	9–18	2
Needle (junction)	18–23	4
Needle cone	23–25	from 4 to d_n_
Needle	25–75	d_n_ = 0.4 or d_n_ = 0.8

**Table 2 bioengineering-10-00448-t002:** Inlet pressure used to achieve different flow rates.

Desired Flow Rate (µL/min)	Needle Diameter (mm)	Kinematic Inlet Pressure (m^2^/s^2^)
50	0.8	6.4
100	0.8	9
500	0.8	20
100	0.4	10

## Data Availability

Data are provided upon request.

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
