# Peer review of "Fabrication of Hydrogel-Based Composite Fibers and Computer Simulation of the Filler Dynamics in the Composite Flow"

_bioengineering, 2023, doi:10.3390/bioengineering10040448_

Round 1

Reviewer 1 Report

This study by Gruhn et al. is quite interesting because the authors produced hydrogel fibres with anisotropic properties by using polymeric microrods as fillers. Casting was used to make the rods, and then they were integrated into an interconnected network made of alginate and GelMA. In general, the concept is intriguing, and both the idea and the manuscript are well-written and simple to understand. The quality of the manuscript could be improved by considering a few of my suggestions, which are relatively minor.

1-The justification for using numerical simulation could be strengthened in some ways. In this reviewer's opinion, it is not entirely clear how numerical simulation contributes to either the experiments or the overall study. The manuscript would be strengthened by including additional elaboration in either the introduction or the results sections.

2-The authors used an amount of alginate and GelAM that was equal to a ratio of 2:1. The question is how they arrived at this ratio. Was that based on a mathematical model or simulation?

3-The authors modelled the microrods as a group of bonded particles in their mathematical representation. Does the roughness of the rods, which is caused by the particles bonding together, have any effect on the hydrodynamics of the fluid?

4- Consider rewriting the caption for Figure 5.

Author Response

This study by Gruhn et al. is quite interesting because the authors produced hydrogel fibres with anisotropic properties by using polymeric microrods as fillers. Casting was used to make the rods, and then they were integrated into an interconnected network made of alginate and GelMA. In general, the concept is intriguing, and both the idea and the manuscript are well-written and simple to understand. The quality of the manuscript could be improved by considering a few of my suggestions, which are relatively minor.

1-The justification for using numerical simulation could be strengthened in some ways. In this reviewer's opinion, it is not entirely clear how numerical simulation contributes to either the experiments or the overall study. The manuscript would be strengthened by including additional elaboration in either the introduction or the results sections.

A1. Thank you very much for your instructive comments. We added some of the considerations in our numerical simulation and their relation to the experimental part to the introduction. The new info can be seen in page 3.

2-The authors used an amount of alginate and GelAM that was equal to a ratio of 2:1. The question is how they arrived at this ratio. Was that based on a mathematical model or simulation?

A2. Thank you very much for your question. This ratio of oxidized Alginate and GelMA was chosen based on the preliminary experiments in the formation of stable fibers after wet spinning. We have tested a lower amount of ADA and observed that the fibers weren’t stable and could not be moved using a tweezer. Furthermore, the viscosity of this solution was measured and used in the numerical simulation.

3-The authors modelled the microrods as a group of bonded particles in their mathematical representation. Does the roughness of the rods, which is caused by the particles bonding together, have any effect on the hydrodynamics of the fluid?

A3. Thank you very much for your instructive comments. In our model w,e assume that the flow field changes only at short distances to spheres, such that the drag force can be applied separately on all particles. This is a reasonable approximation for the given solvent. We have added this comment at page 10 after Eq. 8.

4- Consider rewriting the caption for Figure 5.

A4. Thank you very much for your instructive comments. We rewrote the caption of figure 5.

Reviewer 2 Report

In their manuscript, the authors studied the dynamics of microrods of two different size classes used as fillers of two different composite fibers (wet-spun fibers and 3D-printed fibers) during extrusion with a syringe. The results of both computational simulations and experimental studies are reported. The authors show that the extrusion process leads to a more random orientation of the microrods in the fibers.

This is an interesting manuscript. I only have the following comments/questions:

1. The introduction part is quite long and could be shortened drastically. Many facts are described that are not directly related to the topic of the paper. Only essential information should be mentioned.

2. In this manuscript, microrods were studied, which were prepared using a 12 mg/mL PLGA solution. The results might depend on the material properties of the microrods – do you have any results with microrods made using different concentrations of PLGA? In addition, it would be interesting to see results for longer microrods using the same system – this would help to compare the results with those of other groups. What is the effect of temperature? You may already have some data or can discuss this.

3. Line 90 and line 125 "( D, L-lactide-co-glycolide)" - Please write "D,L" in small capital letters and "co" in italics.

4. Line 98 " In this study, we show the fabrication of composite fibers containing long microrods." The microrods examined in this manuscript are not "long" but rather short compared to those examined in other papers.

5. Line 127 "calcium chloride dehydrate" - please correct: "calcium chloride dihydrate". Or do you mean "calcium chloride anhydrous"? Also, please write more precisely, "CaCl2.2H2O" or "CaCl2", otherwise the concentrations given later are unclear, e.g. in line 165/166 (and also line 316/317) "CaCl2 solution … with 2 and 4.5 wt%"

6. The references are not uniformly cited; e.g. sometimes all words in the titles are capitalized at the beginning, sometimes not.

7. The captions in the Figures are difficult to read. The letter size should be increased. Also the overall size of some Figures could be increased, e.g. the size of Figs 8, 9 and 10.

I think this manuscript is a valuable contribution. The results are new and scientifically sound. They are clearly presented and relevant for the readership of “Bioengineering”.

Reviewer 3 Report

The work presented by Thomas Gruhn and co-workers reports the fabrication of hydrogel-based fibers by wet spinning and 3D extrusion printing. The fibers are formed by interpenetrated networks of oxidized alginate (ADA) and methacrylated gelatin (GelMA) with and without PLGA microrods fillers. Besides, a computational simulation study was carried out to explain the dynamics of the PLGA fillers in the composite flow. I consider the work is accurate for publication in Bioengineering after addressing the following minor revisions:

1) The text needs to be double-checked, to correct the following typing issues:

ml and µl should be changed by mL and µL, with L in capital letter. This needs to be corrected in the text and in the legends of the figures.

In line 226, subscripts in dn needs to be corrected.

In line 334, replace micorods by microrods.

In line 395, replace filametns by filaments.

2) In section 2.3, was PVA spin-coated before or after the PLGA? This information should be added.

3) In section 2.6, what fluorescence probe was used for the fluorescence microscopy measurements?

4) I suggest merging the figures 2 and 3 in only one figure.

5) The experimental technique employed for taking the pictures shown in figures 5A and 5B needs to be mentioned in the caption of the figure.

6) In the Figure 6, what is the difference between the pictures A and B, and between the pictures C and D? If both belong to the same sample, I suggest selecting only one figure for each filler length as the scale bar is the same for both. Besides, what do red arrows mean? It should be mentioned in the caption of the figure, as well as the experimental microscopy technique employed for taking the pictures.

7) In the Figure 7, I suggest using two colors that highlight more the difference between both, as well as using a different color for the error bars of each sample.

8) In line 359, what does ‘k’ mean? I imagine it refers to m in equation 3, and it should appear with the same letter to compare the results. Moreover, it is confusing with the letter ‘k’ used in the equation 9.

9) In the Figure 8, what fiber length does the figure b refer to? I suggest placing the figure b overlapped just with only one of the figures shown in A or in B.

10) What does it means Vx in Figure 10?

11) Author Contributions section: Only the initials of the authors should appear.

12) Data Availability Statement: please replace the text by your data details.

13) The bibliography needs to be revised to accomplish with the specific format of the bioengineering journal: Author 1, A.B.; Author 2, C.D. Title of the article. Abbreviated Journal Name Year, Volume, page range.
